

# A retrospective study using machine learning to develop predictive model to identify rotavirus-associated acute gastroenteritis in children

Sourav Paul[1,*], Minhazur Rahman[2,*], Anutee Dolley[3], Kasturi Saikia[3], Chongtham Shyamsunder Singh[4], Arifullah Mohammed[5], Ghazala Muteeb[6], Rosy Sarmah[7] and Nima D. Namsa[3]

[1] Department of Biotechnology, National Institute of Technology, Durgapur, West Bengal, India
[2] Department of Computer Science and Engineering, Tezpur University, Tezpur, Napaam, Assam, India
[3] Department of Molecular Biology and Biotechnology, Tezpur University, Tezpur, Napaam, Assam, India
[4] Department of Paediatrics, Regional Institute of Medical Sciences, Imphal, Manipur, India
[5] Department of Agriculture Science, Faculty of Agro-based Industry, Universiti Malaysia Kelantan, Kelantan, Malaysia
[6] Department of Nursing, College of Applied Medical Science, King Faisal University, Al-Ahsa, Saudi Arabia
[7] Department of Computer Science and Engineering, Tezpur University, Napaam, Assam, India
[*] These authors contributed equally to this work.

Corresponding authors
Rosy Sarmah, rosy8@tezu.ernet.in
Nima D. Namsa,
namsa@tezu.ernet.in,
ndnamsa12@gmail.com

## ABSTRACT

**Background**. Rotavirus is the leading cause of severe dehydrating diarrhea in children under 5 years worldwide. Timely diagnosis is critical, but access to confirmatory testing is limited in hospital settings. Machine learning (ML) models have shown promising potential in supporting symptom-based diagnosis of several diseases in resource-limited settings.

**Objectives**. This study aims to develop a machine-learning predictive model integrated with multiple sources of clinical parameters specific to rotavirus infection without relying on laboratory tests.

**Methods**. A clinical dataset of 509 children was collected in collaboration with the Regional Institute of Medical Sciences, Imphal, India. The clinical symptoms included diarrhea and its duration, number of stool episodes per day, fever, vomiting and its duration, number of vomiting episodes per day, temperature and dehydration. Correlation analysis is performed to check the feature-feature and feature-outcome collinearity. Feature selection using ANOVA *F* test is carried out to find the feature importance values and finally obtain the reduced feature subset. Seven supervised learning models were tested and compared viz., support vector machine (SVM), K-nearest neighbor (KNN), naive Bayes (NB), logistic regression (Log_R) , random forest (RF), decision tree (DT), and XGBoost (XGB). A comparison of the performances of the seven models using the classification results obtained. The performance of the models was evaluated based on accuracy, precision, recall, specificity, F1 score, macro F1, F2, and receiver operator characteristic curve.

**Results**. The seven ML models were exhaustively experimented on our dataset and compared based on eight evaluation scores which are accuracy, precision, recall, specificity, F1 score, F2 score, macro F1 score, and AUC values computed. We observed that when the seven ML models were applied, RF performed the best with an accuracy

of 81.4%, F1 score of 86.9%, macro F1-score of 77.3%, F2 score of 86.5% and area under the curve (AUC) of 89%.

**Conclusions**. The machine learning models can contribute to predicting symptom-based diagnosis of rotavirus-associated acute gastroenteritis in children, especially in resource-limited settings. Further validation of the models using a large dataset is needed for predicting pediatric diarrheic populations with optimum sensitivity and specificity.

# INTRODUCTION

Diarrheal disease is the second leading cause of mortality and morbidity in children worldwide (*WHO, 2024*). Rotavirus infection is a major cause of acute gastroenteritis (AGE) in children under 5 years of age worldwide. Rotavirus, a major enteric viral pathogen associated with diarrheal disease, is responsible for more than 500,000 deaths per year and is estimated to cause approximately 36% of hospitalizations in children globally (*Crawford et al., 2017*; *Kotloff et al., 2017*). Rotavirus infection continues to pose a major public health burden in children of low-income countries despite the availability of routine rotavirus vaccine immunization (*Aliabadi et al., 2015*). For example, rotavirus was associated with an estimated annual 11.37 million episodes of AGE in children in India, requiring 3.27 million outpatient visits and 872,000 inpatient admissions, resulting in a total direct expenditure of 10.37 billion Indian rupees per year (*Giri et al., 2019*; *John et al., 2014*). The introduction of oral rotavirus vaccines has substantially reduced the severity and the burden of rotavirus-associated AGE worldwide (*Zaman et al., 2010*). However, the mortality and morbidity of rotavirus-led infection are still a major challenge in low- and middle-income countries despite rotavirus vaccine introduction and relatively high coverage of vaccination (*Cunliffe et al., 2014*). Several African countries have conducted studies to investigate the impact of routine rotavirus vaccination, but the results revealed that rotavirus-associated deaths and hospitalizations remain unchanged within 2–3 years post-vaccine introduction (*Varghese, Kang & Steele, 2022*; *Otieno et al., 2020*). The moderate efficacy of oral rotavirus vaccines in developing countries is well recognized and likely due to a multifaceted attribute, including genetic background, malnutrition, comorbidities, or environmental enteropathy (*Lee, 2021*). The moderate rotavirus vaccine effectiveness in children of developing countries may be addressed by improving vaccine coverage and booster doses and developing a next-generation vaccine using the recently described plasmid-based reverse genetics system for rotavirus (*Lee, 2021*; *Desselberger, 2020*).

Rotavirus infection spreads easily *via* the fecal-oral route and causes acute watery diarrhea, fever, nausea, vomiting, and abdominal pain. If untreated, severe rotavirus infection can lead to severe dehydration and even death in children. Therefore, timely

diagnosis and management of rotavirus infection are critical, especially in children under 5 years, where the disease burden is the highest (*Kotloff et al., 2017*; *Iturriza-Go'mara, Kang & Gray, 2004*). Health professionals often diagnose rotavirus based on symptoms and physical examination of the afflicted children. The clinical presentation and stool characteristics of rotavirus-associated diarrhea are often non-specific, and other pathogens may cause similar illnesses. Therefore, confirmation of diarrheal stools requires laboratory testing. Current diagnosis of rotavirus infection uses an enzyme immunoassay (EIA) to detect group A-specific VP6 antigen in the feces of diarrheal children. The performance of the commonly used EIA is considered satisfactory but requires specific reagents and equipment. Further confirmation tests are done by extracting double-stranded RNA of rotavirus from the stool samples and reverse transcription polymerase chain reaction (RT-PCR) and sequencing of VP7 or VP4 genes. Despite the high specificity and sensitivity of RT-PCR, this gold-standard method demands skilled manpower, reagents, and associated equipment, thereby limiting the use of RT-PCR in research institutions and well-equipped hospital establishments to detect the genetic material of rotavirus (*Iturriza-Go'mara, Kang & Gray, 2004*; *Malik et al., 2013*). Hence, routine EIA and RT-PCR testing methods for rotavirus are not accessible in many resource-limited settings where the majority of rotavirus infections are reported.

Machine learning integration into the healthcare sector has the potential to revolutionize medical diagnostics, treatment, and clinical laboratory testing. The artificial intelligence-based machine and deep learning predictive (ML) models have been successfully applied in healthcare sectors for the diagnosis of many diseases, including gastrointestinal disease (*Owasis et al., 2019*), diabetes (*Tigga & Garg, 2020*), cervical cancer (*Ijaz, Attique & Son, 2020*), and coronary artery disease (*Ozbilgin, Kurnaz & Aydın, 2023*). To better measure the effectiveness of routine vaccination and rotavirus-associated disease burden in developing countries, including India, cost-effective and field-deployable rapid diagnostic testing kits are crucial. Currently, there is limited availability of rapid diagnostic test kits for rotavirus disease. There is limited work done on the development of ML-assisted predictive models for the diagnosis of rotavirus-led diarrhea in the pediatric population (*Kananura, 2022*). Machine learning (ML) involves training algorithms on datasets to create mathematical models that can make predictions of many diseases, including cancer (*Kourou et al., 2014*). ML models using clinical data such as symptoms, biomarkers, and demographics have proved useful in identifying rare inherited diseases (*Gomes & Ashley, 2023*). Analysis of proteomics data with the help of ML has led to the identification of biomarkers for alcoholic liver disease, Alzheimer's disease, and Parkinson's disease (*Mann et al., 2021*). Hence, ML or artificial intelligence (AI)-assisted biological data processing can contribute as an alternative, low-cost, and accessible predictive method complementing the requirements of laboratory diagnosis of many human diseases.

A limited investigation has reported the potential application of ML-based predictive models for the diagnosis of pediatric diarrhea. A substantial negative association was found between the rate/coverage of pentavalent vaccination and the prevalence of diarrhea among children living in rural Uganda (*Kananura, 2022*). Infectious disease outbreaks pertaining to diarrhea are a leading cause of morbidity and mortality in South Africa. The climate

variables such as precipitation, humidity, evaporation, and temperature were correlated with diarrhea outbreaks among children in South Africa (*Abdullahi, Nitschke & Sweijd, 2022*). Machine learning algorithms have been able to specifically predict *Clostridioides difficile-associated* infectious diarrhea in hospitalized patients (*Panchavati et al., 2022*). Similarly, ML models have been used to efficiently predict positive cases of waterborne diseases such as typhoid and malaria (*Hussain et al., 2023*). Several supervised learning-based ML techniques have been applied in medical research for developing predictive and diagnostic models for various diseases (*Jia et al., 2019*; *Fuhad et al., 2020*; *Guo et al., 2020*; *Ul Abideen et al., 2020*). Rotavirus infection in children of low- and middle-income countries is still a major pediatric health burden, and the introduction of rotavirus vaccines has substantially decreased the severity of rotavirus-associated AGE (*Aliabadi et al., 2015*). In practice, the symptoms described by patients, physical examinations performed by physicians, and laboratory test results are generally needed to evaluate a patient's status and diagnose a specific disease. However, little research has been conducted into the predictive power and accuracy that can be achieved using clinical symptom data alone for the diagnosis of specific diseases. However, little research has been conducted into the predictive power and accuracy that can be achieved using only clinical symptom data for the diagnosis of specific diseases. Furthermore, there is a lack of research on the development and validation of such ML-assisted prediction of rotavirus-led diarrhea in the pediatric population (*Kananura, 2022*).

Therefore, the purpose of this study is to develop predictive models that physicians can use to make decisions in hospital settings based on ML using clinical symptoms. We will then validate our model through a comparison of its predictions with the diagnoses of physicians. We developed a machine learning model for diagnostic prediction integrating multiple data sources by utilizing clinical parameters collected in collaboration with the Regional Institute of Medical Sciences, Imphal, India, for pediatric rotavirus diarrhea. This study employed seven supervised machine learning models, including SVM, KNN, naive Bayes, logistic regression, random forest, decision tree, and XGBoost.

## MATERIALS AND METHODS

### Data collection and preprocessing

The seven clinical symptoms such as diarrhea duration, maximum number of stool episodes per day, vomiting duration per day, vomiting episodes per day, dehydration, fever, and temperature pertaining to 509 diarrheal children who visited the Pediatric Department of the Regional Institute of Medical Sciences (RIMS), Imphal, India from December 2015 to March 2019, were retrospectively analyzed. The datasets were utilized for training and validation of machine learning predictive models (Table 1) (*Vesikari et al., 1984*). The raw data associated with the study are provided as a Supplementary File. The collected data also included demographic attributes such as the name of the patient, age, sex, vaccination status (Rotavirus vaccines: Rotarix & Rotateq), residential address, family occupation, and family status. Written informed consent was obtained from the study participants, and this study was approved by the Tezpur University Human Ethical Committee ((Number

**Table 1   Summary of clinical dataset that was employed in training and testing of machine learning models.** The seven parameters of the clinical symptoms of diarrheic children who visited the Pediatric Department, Regional Institute of Medical Sciences, Imphal, India from December 2015 to March 2019 were retrospectively analyzed. The datasets were utilized for training and validation of machine learning predictive models.

| Sl. no | Attributes | Missing values | Values (min–max) | Type |
|---|---|---|---|---|
| Feature 0. | Vomiting episodes per day | 0 | 0–10 | Numerical |
| Feature 1. | Vomiting duration (days) | 0 | 0–3 | Numerical |
| Feature 2. | Diarrhea duration (days) | 0 | 1–20 | Numerical |
| Feature 3. | Fever | 0 | Positive–Negative | Categorical |
| Feature 4. | Maximum number of stools (days) | 0 | 0–20 | Numerical |
| Feature 5. | Dehydration | 0 | 1–3 | Numerical |
| Feature 6. | Temperature | 0 | Mild–Moderate–Normal | Categorical |
| Predictive feature | Rotavirus (+ve/-ve) by ELISA | 0 | Positive–Negative | Categorical |

DoRD/TUEC/10-14/2017/4(b)). The dataset was pre-processed by encoding the data into numeric values and removing duplicate entries. Predictive symptoms such as fever, dehydration, diarrhea, and number of stools per day were the most important features obtained according to correlation feature ranking. Of the data collected from 509 diarrheic children, stool samples of 365 diarrheic children tested positive for the presence of rotavirus VP6 antigen using group A-specific anti-VP6 antibody in an ELISA (*Devi et al., 2022*). These samples were confirmed by amplification of VP7 or VP4 genes using the polymerase chain reaction (PCR) followed by sequencing. To develop ML-based predictive methods, models were trained using 80% of the dataset and tested using the remaining 20% data (Table 1). First, data was preprocessed by handling missing values and assessing correlations among various features. Subsequently, seven classification models are applied. Their performances were evaluated based on 20% testing dataset using prediction accuracy, precision, recall, specificity, F1 score, F2 score, macro F1 score, and ROC curves were plotted, and the AUC value for each algorithm was computed. The overall workflow is shown schematically in Fig. 1.

## Feature selection

In this work, we performed a feature correlation analysis on the dataset to understand the underlying relationships among the various features, determine the interdependencies, and assess the effect of each feature on the target feature (*Gopika & ME A.M.K, 2018*). This analysis also helps to understand the redundant features, multicollinearity, and the correlation between different features.

We have also used the filter-based feature selection (FBFS) technique, specifically the ANOVA-*F* test, to identify the most important clinical features from the dataset. FBFS techniques use statistical methods such as similarity, variance, dependence, information, correlation, and distance to indicate the relationship between the input and the target features. In the ANOVA *F*-test, each feature is compared to the target to find any statistically significant relationship between them (*Mishra et al., 2019*). By using correlation analysis and the ANOVA *F* test as a combined approach, we gain insight into both relationships and statistically significant differences between the features.

## Frame Work For Disease Diagnosis

**Figure 1  A schematic framework of machine learning predictive models.** A dataset of 509 children with diarrheic symptoms in the ratio of 80:20 was used for training and testing the seven supervised machine learning algorithms. The best-performing models were selected and developed as predictive diagnostic models for rotavirus diarrhea in the pediatric population.

## Machine learning models

In this study, we used seven supervised machine-learning algorithms to predict positive cases of rotavirus infection. Based on the patient's symptoms, the prediction was made using SVM, KNN, Log_R, RF, DT, NB, and XGB. The models are briefly described below.

## Support vector machine

SVM is utilized for classification and regression to effectively handle non-linear data with high accuracy and speed, especially when variable relationships are unclear (*Hearst et al., 1998*). SVM has been effectively used in bioinformatics, especially in disease diagnosis and prediction (*Maglogiannis, Loukis & Zafiropoulos, 2009*). *Srivastava, Kumar & Singh (2022)*

proposed a hybrid disease diagnosis framework for diabetes prediction using LS-SVM. Similarly, *Elsedimy, AboHashish & Algarni (2024)* proposed a heart disease detection model based on the quantum-behaved particle swarm optimization (QPSO) algorithm and SVM to analyze and predict heart disease risk.

### K-nearest neighbor

KNN classifies cases based on distance functions, adjustable through the 'K' parameter, making it flexible for classification and regression (*Xing & Bei, 2020*). *Chandel et al. (2016)* carried out experimental research using the RapidMiner tool, and the findings indicate that the K-nearest neighbor method outperforms naive Bayes in detecting thyroid disease. *Alanazi (2022)* proposed a method for identifying and predicting chronic diseases using machine learning algorithms like convolutional neural network (CNN) and KNN, showcasing higher accuracy compared to other algorithms.

### Logistic regression

Log_R is ideal for binary classification. It calculates probabilities using a logistic function and evaluates multiple independent variables (*Nick & Campbell, 2007*). *Ambrish et al. (2022)* discussed a logistic regression model for classifying heart disease. The model achieved an accuracy of 87.02%.

### Decision tree

DT, which splits data based on feature values, is easy to understand and quick to use but it is prone to overfitting unless pruned (*Costa & Pedreira, 2023*). *Tanner et al. (2008)* demonstrated the effectiveness of decision algorithms in diagnosing dengue and forecasting severe disease, aiding in patient care and public health efforts.

### Random forest

RF is an ensemble of decision trees that enhances classification accuracy and is robust against noise, making it suitable for large datasets (*Breiman, 2001*). *Khalilia, Chakraborty & Popescu (2011)* observed that the RF ensemble learning method outperformed other techniques in predicting disease risks based on patients' medical diagnosis history, achieving an average AUC of 88.79%. *Wang et al. (2020)* analyzed the clinical characteristics of COVID-19 patients using a random forest algorithm to predict patient outcomes and identify potential risk factors for mortality.

### XGBoost

XGBoost, or eXtreme Gradient Boosting, is known for its high performance in boosting prediction accuracy and is particularly effective for structured, complex datasets (*Chen & Guestrin, 2016*). *Sankar et al. (2022)* proposed an XGBoost algorithm for accurate thyroid disease prediction, outperforming traditional methods with a 2% increase in accuracy, this emphasizes the importance of early detection in preventing adverse health conditions.

### Naive Bayes

NB is a probabilistic classifier based on Bayes' theorem that efficiently handles large datasets with many features and is widely used in medical diagnostics (*Mathur & Joshi,*

*2019*). *Trihartati & Adi (2016)* present a study on identifying tuberculosis in humans using the naïve Bayesian method, achieving an average accuracy of 85.95%. Each of the methods discussed above varies in flexibility and computational speed, from fast to moderate to slow, and their practical applications are well-documented in respective software manuals and scientific literature.

## Performance evaluation metrics

The results of the classification models were validated using 20% testing dataset and seven validation indices: accuracy, precision, recall, specificity, F1, macro F1, F2 scores, and the receiver operating characteristic curve (ROC) (*Lever, Krzywinski & Altman, 2016*). The evaluation metrics, calculated from confusion matrices, are detailed in Table 2. Accuracy represents the number of correctly classified samples among all samples produced by the model. Precision reflects the proportion of actual positive samples among all retrieved positive samples, serving as a measure of quality. Recall measures the completeness of the model, represented by the fraction of actual positive samples that were identified as positive. The F1 score indicates the model's reliability and is calculated as the harmonic mean of precision and recall. The F2 score places greater emphasis on recall than on precision and ranges from 0 to 1, with higher values indicating better performance in terms of recall. The macro F1 score computes the unweighted mean of all F1 scores for each class (*Opitz, 2022*). The ROC technique is widely used for disease classification. It is a probability curve where classification performance is assessed by the area under the ROC curve (AUC), which measures the model's ability to distinguish between classes (*Ma & Huang, 2007*). The area under the ROC curve (AUC) graphically represents the performance of binary classification models in predicting positive or negative classes of diseased and non-diseased individuals. The performance evaluation metrics were computed using the predicted labels and the actual labels of the testing samples.

## RESULTS

In this study, we present an in-depth analysis of seven selected machine learning models and explore their potential for predicting rotavirus infection in children from low- and middle-income countries, where the cases of rotavirus-associated AGE remain high compared to those in developed countries. The dataset included 509 diarrheal children who visited the Pediatric Department of the RIMS, Imphal, India, from December 2015 to March 2019. According to our previous study, the RIMS serves as a catchment care hospital that covers patients from many districts in Manipur state, India. Of the 509 diarrheal children, stool samples from 365 diarrheal children were confirmed by ELISA for the presence of rotavirus antigen using group A-specific anti-VP6 antibodies. The remaining 144 stool samples did not have rotavirus infection despite the symptoms associated with the diarrheal children (Table 1). These samples were confirmed by the polymerase chain reaction (PCR) amplification of VP7 or VP4 genes and sequencing (*Devi et al., 2022*). Of the 509 preprocessed data samples, 80% were utilized for training and 20% for testing during model development and evaluation.
**Table 2  Performance measurement matrices of machine learning models.** Description and formulation of performance metrices of eight different supervised machine learning algorithms.

| Metrics | Description | Formula |
|---|---|---|
| Accuracy | Percentage of satisfactory cases among all occurrences | $\frac{TP+TN}{TP+FP+FN+TN}$ |
| Precision | Percentage of true positive predictions among all positive cases | $\frac{TP}{TP+FP}$ |
| Recall (Sensitivity) | Proportion of all true positive cases among all actual positive cases | $\frac{TP}{TP+FN}$ |
| Specificity | Proportion of true negative cases removed or detected among all negative cases | $\frac{TN}{TN+FP}$ |
| F1-score | The harmonic mean of Precision and recall | $2 * \frac{Precision*Recall}{Precision+Recall}$ |
| Macro-F1-score | This is the unweighted mean of the f1 score. | $\frac{F1_{class\_1}+F1_{class\_2}+...+F1_{class\_n}}{n}$ |
| F2-score | This emphasizes recall more than precision. This is more significant when false negative is more important than false positive. | $5 * \frac{Precision*Recall}{4*(Precision+Recall)}$ |

In Table 1, we summarize the dataset concerning clinical features of diarrheic children. Computed correlation values between the different medical features and rotavirus (the target disease) are shown in Fig. 2. We can see in the figure that of the seven features of our dataset, five features have a positive correlation with the decision feature, *i.e.,* 'rotavirus.'. The features 'vomiting episodes per day', 'vomiting duration in days', 'fever', 'maximum number of stools (days)', and 'temperature' have correlation values of 0.087, 0.036, 0.069, 0.06, and 0.039, respectively with 'rotavirus', indicating a positive correlation. In contrast, two features 'diarrhea duration (days)' and 'dehydration' are negatively correlated with the values of −0.047 and −0.027, respectively. These features may be considered important decision factors. According to published research findings, the occurrence of fever and vomiting are commonly associated with rotavirus diarrhea compared to non-rotavirus diarrhea in children (*Salim et al., 2014*). Our correlation analysis between each of the features also shows two of the features, "fever" and "temperature," having a high correlation of 0.67 (Fig. 2).

The results of the ANOVA-*F* test showed that 'fever' (with a *F*-test score of 4.69), 'vomiting episodes per day' (*F*-test score of 3.64), and 'temperature' (1.514) are the most important features for predicting rotavirus infection related to the target (Fig. 3). The scores of the other features are given as 'vomiting duration in days' (0.75), 'diarrhea duration in days' (0.44), 'maximum number of stools per day' (0.36), and 'dehydration' (0.12) when related to the target. Looking at the importance values of the features, we can observe a similarity with the correlation results listed above; in most cases, the features related to fever, vomiting episodes per day, temperature, vomiting duration in days, maximum number of stools (days), and diarrhea duration have a significant influence in the prediction of rotavirus disease. The features identified using the ANOVA-*F* test are also listed as potential factors for diagnosing rotavirus, as reported previously (*Nnukwu et al., 2017*). *Nnukwu et al. (2017)* reported that rotavirus disease usually starts with the onset of vomiting and fever, followed by mild to frequently profuse diarrhea resulting in
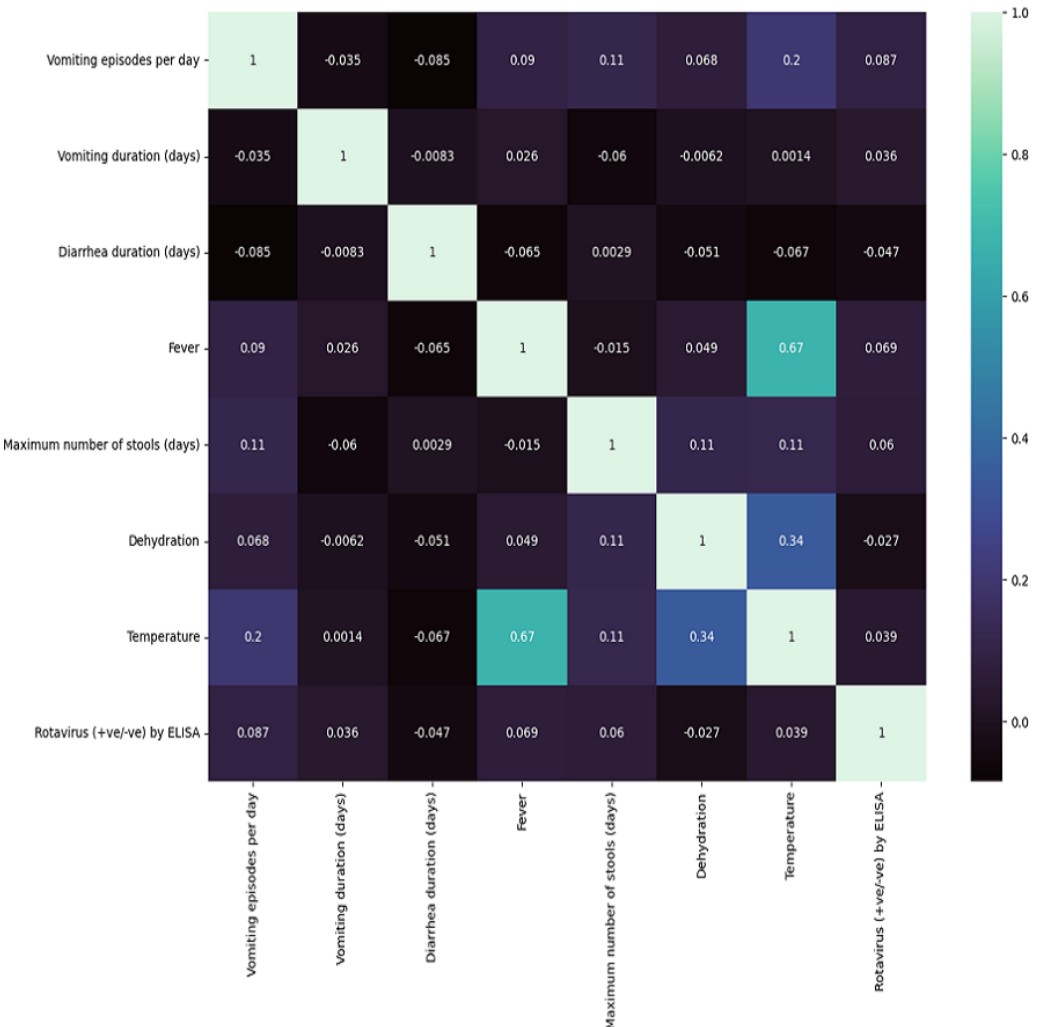

**Figure 2 Correlation matrix features .** Seven clinical parameters that are presented by diarrheic children
($n = 509$) were utilized to find the correlation of the features with the outcome of rotavirus disease.

dehydration, imbalance in electrolytes, and death. Rotavirus infection in children with
diarrhoea is correlated as reported previously, while the children with no presentation
of diarrhoea is tested negative for rotavirus infection (*Nnukwu et al., 2017*). ANOVA *F*
test feature ranking showed high scores for both fever and temperature. Hence, we have
performed our analysis by considering both the features. We have performed additional
analysis to re-evaluate the performance of ML models removing either fever or temperature
to avoid collinearity. We observed a slight increase or decrease in the performance of some
ML models, while the performance of some ML models remained unchanged both at
accuracy and AUC under condition when either of the features (*i.e.,* fever or temperature)
were retained or removed (Figs. S2 and S3, Table S2).

To attain the goal of identifying the potential diagnostic factors and analyzing the
influence of feature selection on the accuracy of classification, we performed the analysis

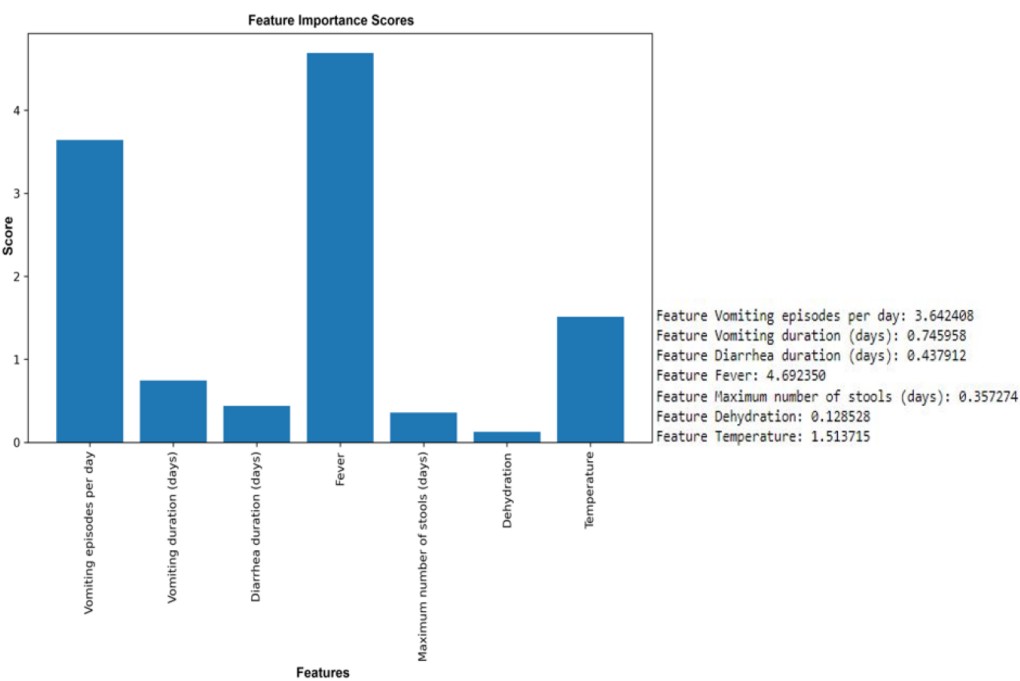

**Figure 3  Feature selection using the ANOVA *F* test.** Top six features except dehydration have been selected for performance measurement and evaluation using seven supervised machine learning algorithms.

using the reduced feature set, omitting the dehydration feature. This set was identified based on the ANOVA-*F* test scores of each clinical feature. All seven classification models, such as RF, NB, DT, KNN, SVM, logistic regression, and XGBoost, were trained on 80% train dataset and 20% testing dataset. The total computational time consumed during the training of prediction models was 1.6714 s (Table S1). Table 3 shows the classification results of the ML models based on 20% testing dataset concerning accuracy, precision, recall, F1 score, macro F1, F2 score, and specificity in predicting rotavirus disease. Of the seven supervised learning models tested, the RF, XGBoost, and DT models showed good performance with an accuracy of 81.4%, 78.4%, and 72.5%, respectively (Table 3). RF also recorded the highest F1 and macro F1 scores. For the other scores of recall, F2, and specificity, RF obtained scores above 85%, showing a good overall performance. In the present study of classification, the RF, XGBoost, and DT exhibited good performance with AUC values of 89%, 84.9%, and 78.6%, respectively (Fig. 4). The classification results in Table 3 indicate that the highest accuracy reported was 81.4% achieved by RF with an F1-score of 0.869 (Table 3).

The confusion matrices are reported on 20% of the test dataset to further evaluate the performance of the best ML models (RF and XGBoost). The row of the matrix corresponds to the actual class, and each column contains the predicted class. The rotavirus positive and negative cases are denoted as 1 and 0, respectively, and the cases are reported in the confusion matrix. XGBoost predicted 18 rotavirus positives as positive cases and 62 rotavirus negatives as negative cases. The 11 negative values were wrongly predicted as

**Table 3  Performance evaluation metrics of machine learning models.** The performance of ML models was measured by evaluation metrics using six feature set.

| Algo. | Accuracy | Precision | Recall | F_macro | F1 score | F2 score | Specificity (%) |
|---|---|---|---|---|---|---|---|
| SVM | 71.57 | 0.7157 | 1 | 0.4171 | 0.8343 | 0.9264 | 100 |
| KNN | 69.6 | 0.714 | 0.959 | 0.4396 | 0.819 | 0.897 | 95.89 |
| Logistic Regression | 71.6 | 0.716 | 1 | 0.42 | 0.834 | 0.926 | 100 |
| Naïve Bayes | 65.7 | 0.702 | 0.904 | 0.422 | 0.790 | 0.854 | 90.41 |
| Decision Tree | 72.5 | 0.7647 | 0.89 | 0.607 | 0.823 | 0.862 | 89.04 |
| Random Forest | 81.4 | 0.875 | 0.86 | 0.773 | 0.869 | 0.865 | 86.3 |
| XGboost | 78.4 | 0.849 | 0.735 | 0.735 | 0.849 | 0.849 | 84.93 |

rotavirus-positive cases, and 11 positive cases were incorrectly predicted as rotavirus-negative (Fig. S1B). On the other hand, both XGBoost and the RF were able to precisely classify 259 and 252 rotavirus-negative samples as negative cases from the training data, respectively (Figs. S1A and S1C). In contrast, the RF model predicted 20 rotavirus-positive cases as positive and 63 rotavirus-negative cases as negative. The 10 negative cases were incorrectly predicted as rotavirus-positive, while nine positive rotavirus cases were wrongly predicted as rotavirus-negative (Fig. S1D).

# DISCUSSION

In healthcare, ML models have shown capability in the diagnosis of diseases based on patient characteristics. In this study, seven different supervised learning-based ML techniques have been applied to develop diagnostic models for pediatric rotavirus diarrhea as an alternative cost-effective method for diagnosing rotavirus disease detection (*Jia et al., 2019*; *Fuhad et al., 2020*; *Guo et al., 2020*; *Ul Abideen et al., 2020*). The datasets included the seven parameters of clinical symptoms such as fever, diarrhea, and its duration, number of stool episodes, vomiting and its duration, number of vomiting episodes, temperature, and dehydration. These parameters pertain to 509 diarrheal children in a rural setting who visited the Pediatric Department, RIMS, Imphal, India, from December 2015 to March 2019 (*Devi et al., 2022*). These seven scoring parameters are included in the Vesikari clinical severity scoring system and are currently recognized as the most accurate system for use in developing country rotavirus vaccine trials (*Vesikari et al., 1984*). The results of several published works implicated the lack of specific symptoms of rotavirus gastroenteritis in either inpatients, outpatients, or age (*Binka et al., 2003*; *Vesikari, Sarkkinen & Maki, 1981*). However, rotavirus-positive children generally have a shorter duration from disease onset to hospital visits (*Senecal et al., 2008*; *Kim et al., 2005*).

We performed correlation analysis to find the relationship between clinical symptoms and their impact on rotavirus disease outcome. Feature scaling could help normalize the data, potentially leading to better overall results for the models in general. However, since medical studies do not advocate any data manipulation (as it might introduce bias), we have not used scaling and have kept all the feature values unchanged. The heat map of the correlation plot showed that fever and temperature exhibited a positive correlation

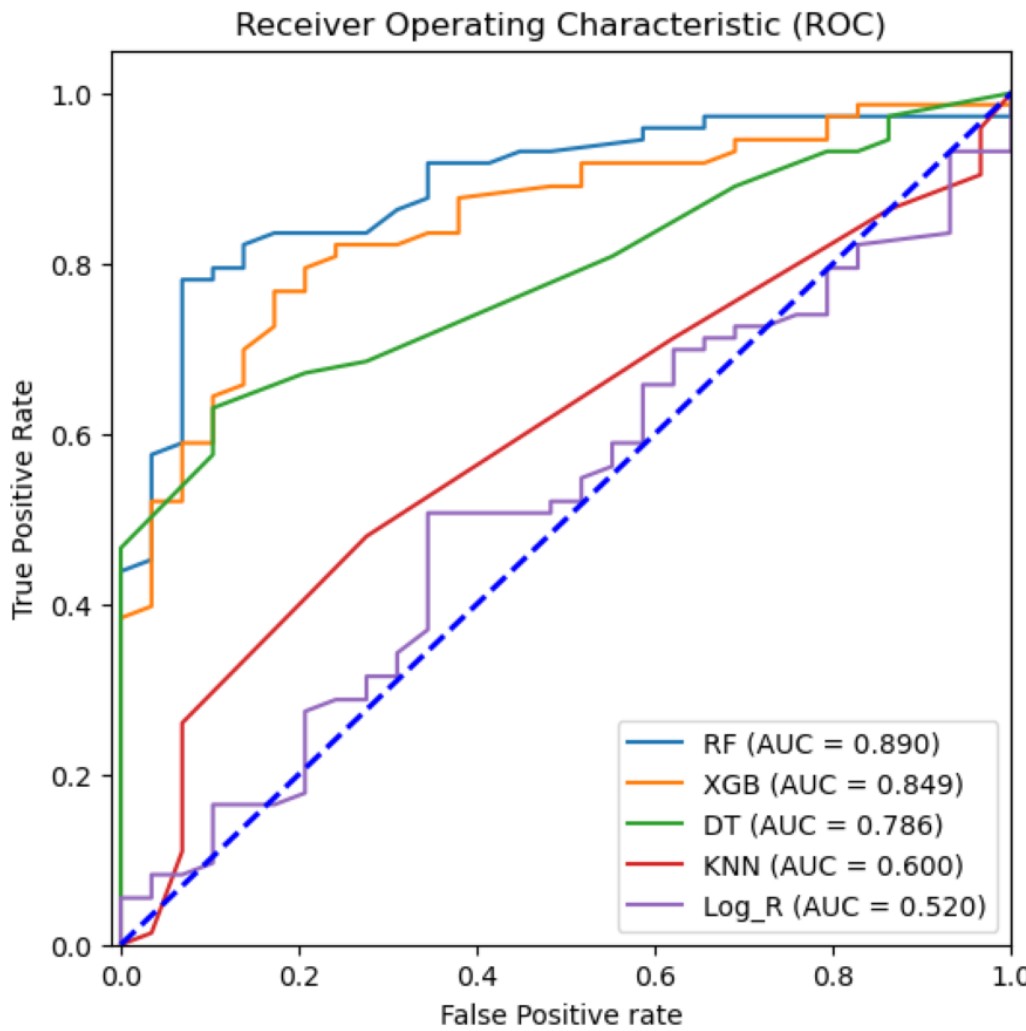

**Figure 4 Machine learning prediction of rotavirus-associated acute gastroenteritis.** Machine learning prediction of rotavirus-associated acute gastroenteritis. Five supervised ML algorithms (RF, XGB, DT, KNN, & Log_R) showed good performance based on the receiver operator characteristic curve (ROC). The ROC curve takes the false-positive rate as the horizontal axis and the true-positive rate as the vertical axis. The horizontal axis represents the proportion of the actual negative instances in the positive class predicted by the classifier to all negative instances. The vertical axis represents the proportion of the actual positive instances in the positive class predicted by the classifier to all positive instances. The area under the curve (AUC) represents the ability of models to differentiate between positive and negative values during prediction.

with the outcome of rotavirus disease. Because the symptoms and duration of diarrhea, vomiting, fever, and dehydration commonly occur with diarrheal illnesses caused by other etiological agents, the findings suggest that there is no combination of symptoms or specific symptoms that confirms that a person has rotavirus infection (*Vesikari et al., 1981*). We performed correlation analysis on the feature space to understand the relationship and interdependencies between features as well as the influence of these features on the target outcome (rotavirus). This analysis revealed that five out of seven features have a positive

correlation with the target. It was also found that there is a high correlation between two features (fever and temperature). Furthermore, the feature selection technique of the ANOVA-*F* test was used to identify the most important features. This analysis ranked both fever and temperature with high scores, showing that both features contribute to the diagnosis of rotavirus disease (*Nnukwu et al., 2017*). Feature based feature selection is an important statistical method of selecting the best feature for performance measurement and evaluation of ML models. Hence, 'dehydration' received a very low score in both our correlation and FBFS analysis and excluded from ML model-based performance measurement. It has been reported that dehydration is found clinically relevant and positively correlated with the rotavirus infection in children with diarrhoea in Nigeria (*Nnukwu et al., 2017*; *Junaid et al., 2011*). The relative performance of seven supervised machine-learning algorithms is helpful in the identification and selection of an appropriate machine-learning algorithm for the prediction of rotavirus disease (*Uddin et al., 2019*). RF, XGBoost, and DT demonstrated good performance with an accuracy of 81.4%, 78.4%, and 72.5%, respectively, for all seven features. RF, XGBoost, and decision tree models predict rotavirus-associated AGE with the precisions of 87.5%, 84.9%, and 76.47%, respectively. The diagnostic ability of ML models has been determined by the confusion matrix and the area under the receiver operating characteristic curve. In this study, RF and XGBoost showed better performance in classifying feature space with AUC values of 89% and 84.9%, respectively. The reason behind the improved accuracy achieved by RF is that it does not require normalization or scaling of the data, avoids overfitting, and is good at discovering patterns from complex medical datasets. In a previous work, a supervised gradient-boosted ML technique was developed to identify predictors of diarrhea and the result showed a test accuracy of 70% in rural and 100% in urban settings in Uganda (*Kananura, 2022*). Rotavirus AGE has a unique clinical presentation profile that remains consistent across regions and economic settings. Studies have found that vomiting and fever accompanying diarrhea differ across studies, possibly based on the age of children, inpatient and outpatient. Vomiting and diarrhea occur in more than 50% of patients with rotavirus. Dehydration is often found in rotavirus-positive participants than in rotavirus-negative subjects (*Nyambat et al., 2009*; *Zaman et al., 2009*). Therefore, when we removed dehydration from the input feature set, there was a negligible increase in the evaluation scores. However, our correlation analysis indicates that most features equally contributed to this dataset, which is the ideal case for a decision tree to work well.

The ML models were tested using a small dataset of clinical parameters collected from 509 diarrheal children who visited RIMS, Manipur, India. The dataset was collected from a single hospital, and the ML model performance may vary using clinical features of different patient populations. In addition to supervised machine learning models, neural network architectures and hyperparameters could be utilized for model optimization and validation. Moreover, as a future direction of work, a meta-ensemble model may be envisioned and exhaustively tested on a larger and geographically diverse rotavirus dataset to further validate the effectiveness of ML models as a low-cost diagnostic tool.

## CONCLUSIONS

This study described the performance of seven different supervised ML methods for the identification of predictors of rotavirus-associated diarrhea in children in resource-limited settings, and the models, especially RF, XGBoost, and DT, showed good performance in predicting rotavirus-led AGE. In conclusion, RF and XGBoost models have the potential for integration into health information systems to enable a low-cost predictive method for diagnosis of rotavirus in low- and middle-income countries in a resource-limited hospital setting from electronic records.

## ACKNOWLEDGEMENTS

Rosy Sarmah and Nima D. Namsa thank the BioNET lab of the Department of Computer Science & Engineering where all the computational work, including design, analysis, implementation, and experimentation, was conducted as well as the Bioinformatics and Computational Biology Centre, Tezpur University for the use of the computing facility.

### Funding

This work was supported by the Deanship of Scientific Research, Vice Presidency for Graduate Studies and Scientific Research, King Faisal University, Saudi Arabia (Grant No. KFU250435). The funders had no role in study design, data collection and analysis, decision to publish, or preparation of the manuscript.

### Grant Disclosures

The following grant information was disclosed by the authors:
Deanship of Scientific Research, Vice Presidency for Graduate Studies and Scientific Research, King Faisal University, Saudi Arabia: Grant No. KFU250435.

### Competing Interests

The authors declare there are no competing interests.

### Author Contributions

- Sourav Paul performed the experiments, analyzed the data, prepared figures and/or tables, authored or reviewed drafts of the article, and approved the final draft.
- Minhazur Rahman performed the experiments, analyzed the data, prepared figures and/or tables, authored or reviewed drafts of the article, and approved the final draft.
- Anutee Dolley performed the experiments, analyzed the data, prepared figures and/or tables, eLISA, and approved the final draft.
- Kasturi Saikia performed the experiments, analyzed the data, prepared figures and/or tables, pCR, and approved the final draft.
- Chongtham Shyamsunder Singh analyzed the data, prepared figures and/or tables, sample collection, and approved the final draft.

- Arifullah Mohammed analyzed the data, authored or reviewed drafts of the article, and approved the final draft.
- Ghazala Muteeb analyzed the data, authored or reviewed drafts of the article, and approved the final draft.
- Rosy Sarmah conceived and designed the experiments, performed the experiments, analyzed the data, authored or reviewed drafts of the article, and approved the final draft.
- Nima D. Namsa conceived and designed the experiments, performed the experiments, analyzed the data, authored or reviewed drafts of the article, and approved the final draft.

## Human Ethics

The following information was supplied relating to ethical approvals (i.e., approving body and any reference numbers):

Tezpur University Human Ethical Committee (number DoRD/TUEC/10-14/2017/4(b)).

## Data Availability

The raw data are available in the Supplementary File.

## Supplemental Information

Supplemental information for this article can be found online at http://dx.doi.org/10.7717/peerj.19025#supplemental-information.

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
