# Peer review of "A retrospective study using machine learning to develop predictive model to identify rotavirus-associated acute gastroenteritis in children"

_PeerJ, doi:10.7717/peerj.19025_

## Round 0.1 · original submission · Major Revisions

Overall, the work by Paul et al can be very useful in improving the accurate diagnosis of AGE. Although this work is promising, the authors should make substantial changes to various aspects of the work before moving forward in the process (please see the reviewers' comments for a more detailed description).

Reviewer 1 made some very useful comments about basic reporting, experimental design, and validity of the findings that should be thoroughly addressed. In addition, major changes should be made in the study’s experimental design, and a substantial effort to improve the validity of the findings. I would also recommend addressing the comments provided by Reviewer 3.

·

Basic reporting

The authors fluently described the necessity (a low-cost screen kit for rotavirus infection in children) and technical plausibility of their study (based on recent R&D in applied machine learning in clinical practice), with adequate citation and background support.
However, the depiction of 8 machine learning methods are confusing, not due to English but poor under-stand of the methods themselves (see the section comments below). For an applied study emphasizing clinical partiality, the authors could discard the depiction machine learning methods all together, and simply state the computation cost (fast, moderate, slow, etc.), model flexibility (linear, non-linear, and ensemble) in one short sub-section. For each method, there is no need to provide theoretical explanation or any formula, but one citation of the original theory (authors do not have to fully understand) and at least cite a tutorial for readers interested in apply the method and replicating the study. If the method was provided as software packages, citing the package manual as a tutorial will suffice. After shrinking the description of 8 Machine Learning methods, authors should add to the method section descriptions of performance metrics (e.g., accuracy, AUC, and macro F1 score), and the reason to calculate correlation coefficients.

Regardless the validity of findings, the result section reported well-formatted tables and figures. Although some expressions in the main text are still confusing or redundant, they are much easier to fix than the 8 methods.

The authors also provided a well-formatted, de-identified raw data sheet which is readily readable by any statistical analysis software, therefore, future researchers should be able to re-run and improve the study with materials so provided, possible with new machine learning methods.

Experimental design

The retrospective case / control study design is intuitively understandable. The upstream bio-sampling and laboratory test seems professional and the data is robust.

However, the downstream execution of machine learning showed two major flaws: (1) the case / control ratios are drastically different among training and testing datasets; and (2) the data split (into training and testing) varied for each of the 8 methods, making the prediction performance incomparable across the methods (see comments on Fig.5).

If possible, could the authors include child demographics (i.e., age at hospital visit, and sex) as additional predictors and check the prediction performance, or argue the reason not to include predictors which are typically used by a case control study?

Validity of the findings

The current findings are invalid due to reasons listed in the previous point (i.e. Experimental design). The authors must at least fix the two major flaws in the preparation of training and testing data, then re-run the machine learning methods to obtain valid findings (i.e, benchmarks of prediction performance).

Additional comments

Comments on the main text
The first paragraph of introduction (63 – 89) laid out the compelling fact in support of machine learning based screening tools with favorable cost – benefit compared to standard laboratory tests, despite the latter’s high detection power.
Introduction to current field of study
Between line 103 and 105, the text phrased that the association between vaccination and diarrhea was found (1) among those who were not vaccinated, or (2) (those who live in communities of) low vaccine coverage. Typically, detecting the association between an exposure (vaccination) and an outcome (diarrhea) requires both exposed (i.e., non-vaccinated) and unexposed (i.e., vaccinated) samples. A better expression maybe: “A substantial negative association was found between (the rate/coverage of) pentavalent vaccination and (the prevalence of) diarrhea among children living in rural Uganda”.

From line 119, the authors have finished the introduction and shifted to describing their own work, starting with one of the aims, therefore I think the text starting from line 121 (before Material and Methods) warrant a paragraph on its own.
In line 123, consider changing “laboratory and clinical” to “clinical” alone, since the manuscript has emphasized the high cost of formal laboratory diagnosis in the beginning, and argued (between line 121 and 123) for a much cheaper, machine learning model based on clinical symptoms alone.
Between line 124 and 127 (starting with “Despite”) to 129, the authors stated another aim, it would be better to move it right after the main aim in line 121 (starting with “However”), consider the following:
However, little research has been conducted into the predictive power and accuracy that can be achieved using clinical symptoms data alone for the diagnosis of specific diseases; in addition, there is a lack of research on the development and validation of such ML-assisted prediction of rotavirus-led diarrhea in pediatric population. Therefore, …
In line 155, consider moving "performance of" into the next sentence, after "and", as "performance of the models were evaluated by …", because one could apply an algorithm but not a performance.
Describing Support Vector Machine
Between line 166 to 174, the supposed description of SVM (Support Vector Machine) unnecessarily turned the emphasis to SVR (support vector regression) instead. For this particular study, the outcome of interest is binary (EIA/RT-PCR positive versus negative), therefore the traditional SVM as a classifier should be the natural choise; SVR as an extension of SVM can analysis quantitative outcome (e.g., blood pressure, fasting glucose level, etc.), but does not fit the requirement of this study.
A better description could focus on the rationale of SVM and its advantage over other methods, such as high robustness due to implicit regularization, and the ability handle a large number of predictors due to the use of kernel, although such advantage is not highlighted here since only a handful of clinical symptoms are used as predictors.
If the mathematical rationale of SVM is difficult to grasp and explain, simply stating what the SVM method can do (i.e., building robust classification models), its advantage over other methods, followed by citing relevant literature (preferably the original theory and a tutorial), are adequate for an application-oriented manuscript.
In this study, citation [22] (a tutorial for SVR, but not SVM) should probably be replaced by [Hearst, M. A., Dumais, S. T., Osuna, E., Platt, J., & Scholkopf, B. (1998). Support vector machines. IEEE Intelligent Systems and their applications, 13(4), 18-28].
Describing K-nearest neighbor
Please revise the description between line 177 and 179, considering the following:
“The prediction is based on the majority votes of neighboring data points near the point of interest (within the space spanned by predictors). For this purpose, it is necessary to measure the closeness among data points, where the Euclidean, Manhattan, and Minkowski distances are considered.”
Line 179-181 did not specify how the choice of k (the number of nearest neighbors) affects the flexibility and robustness of a KNN. To my knowledge, larger k (bigger voting neighborhood) favors robustness (prevent overfiiting) over flexibility (promote accuracy in the training data), while smaller k does the opposite.
Searching the k that balances robustness and flexibility can be a tedious task on its own, could you report the final k used in the study, and how did you find it (could it be the software package self-determined it via cross-validation)?
Between line 185 and 186 the statement of KNN being used for predicting data with overfitting is confusing. It is not unusual that sophisticated models overfit a small training sample (reaching 100% accuracy), but it is rare to say a data itself came with an overfitting property. Did the authors actually mean the KNN is capable to analyze data with more predictors than samples (e.g., more than a million-nucleotide polymorphism typed form several thousands of people), which indeed could result in over-fitting if one naively builds a classification model with too many predictors?
Line 183-186: please change “euclidian” to “Euclidean”. The subscripts of two instance are ranged from 1 to n are meant to index predictors, but n is typically, and has been used by the manuscript, to represent sample size, so please consider using a different symbol to specify the number of predictors, or simply use the actual number, which should have been 7 (but 6 was given); similarly, y is typically reserved to represent the outcome (EIA/RT-PCR positive versus negative), please choose a different symbol. As an example, one could use “m” for the number of predictors, “a” for the first, and “b” for the second instance, so the two data points would be like (a1, a2…, am) and (b1, b2, … bm), or, one could use 2-tiered subscript for x, so the two would be (x11, x12, … x1m) and (x21, x22, … x2m).
In line 190, the formula tried to specify the actual number of predictors which should have be 7 but 6 was provided; in addition, the symbol “j” indexing the predictors is misleading because the subscripts of two instance x and y was “i”. Typically, letter “i” indices samples and “j” indices predictors, please change “n” to “m”, “i” to “j”, and “x”, “y” to whatever symbol you decided to represent the two instances.
Since line 180 has stated that three distance measures were used (Euclidean, Manhattan, and Minkowski), pleased add the formal description of the latter two; or, if the study only report the best performing distance measure (i.e., Euclidean maybe) for a final KNN, please state so in the previous paragraph (that we chose Euclidean distance which is best performing).
Describing Linear Regression
Linear indeed predicts a quantitative variable by relating the said quantitative variable (i.e., an outcome) with one or more explanatory variables (i.e., predictors). However, the authors are yet to justify the use of linear regression since the outcome of interest is a qualitative, that is, EIA/RT-PCR positive or negative. An appropriate quantitative outcome would be the normalized PCR readings, or some continuous scale representing the amount of rotavirus DNA enriched by PCR.
Although by force, one could use linear regression to model and predict binary outcome coded as 0=negative and 1=positive, but not without specifying a threshold T, so predicted value greater than threshold T is reported positive, otherwise negative. The choice of threshold T affect the sensitively, specificity, accuracy, and F1 score. A common choice of threshold can be the estimated intercept (i.e., alpha_0 fitted by the linear regression software).
Linear regression is in fact the simplest parametric model used in the study, which actually is the least likely to overfit, fastest to build the model. The other method can be as fast and as linear regression is the non-parametric KNN (non-parametric means, as the author stated, it does not require fitting a model), but KNN can still be much slower depending on the type of distance metric used.
Regarding the outliers, all algorithms are affected. If one of the algorithms in the study is known to be more resistant to outliers, the authors should highlight it.
If the above concern has never been addressed, the authors may consider removing linear regression from the methods and results all together.
Describing Logistic Regression
The description of logistic regression was imprecise. Consider the following example:
Logistic regression models the probability of infection or EIA/RT-PCR test positive (denoted by y = 1) as the sigmoid transformation of the linear combination of predictors (x1, … xm, that is, 7 clinical symptoms). Formally, it writes
Pr(y = 1) = g(α0 + α1x1 + … + αmxm), m = 7
where the transformation g is sigmoid function g(µ) = 1 / (1 + e-µ).
After fitting the model with training samples, it can make prediction as probabilities of rotavirus infection based on 7 symptoms of the testing samples. Unlike linear regression where one has to decide a threshold to make a positive call, most logistic regression software package will report positive when the predicted probability is greater than 0.5 (i.e., more than 50% change of being infected).
Decision Tree and Random Forest
The Decision tree is indeed quite easy to interpret due to the resulting model having an if-else like logic. However, fitting a decision tree is not as fast and simple as fitting a linear regression or logistic regression model. The decision tree is capable of modeling non-linear association between the predictors and the outcome, for example, the relation between one predictor and the outcome may reverse (from promoting to demoting) depending on the value of another predictor.
In line 217 the author said the model (decision tree) is well suited to classify unknown datasets, does it actually mean data with missing values? Indeed, the decision tree can treat “missing” as one of the choices, which can be advantageous if the study data has substantial missing values. Nonetheless, please change the expression from “unknown data” to “missing values”.
In line 221, the authors stated that the trees in a Random Forest are independent, but actually they are not, despite each tree capturing different aspects of the relationship between an outcome and multiple predictors. The reason of non-independence is because the trees were built from bootstrap samples of the same training data.
In line 222, as the authors stated that the random forest is more capable of classifying complex outcome than linear algorithms, but they did not specify what are linear algorithms (linear regression and logistic regression), and the reason why random forest is more capable (due to the decision trees are capable of modeling non-linear relationship between the outcome and the predictors).
In line 223, the authors said the random forest can work at a relatively high classification speed, does it actually mean a relatively high speed in model fitting (instead of the prediction after model fitting)? Also, relative to what method?
Describing XGBoost
(I will refrain from commenting on XGBoost because I am not well practiced in this method).
Describing Naive Bayes
In line 235, I think the author correctly stated the advantage of Naive Bayes, that it is capable of classification task based on high dimension data. After all, the method naïvely assumes independence between predicators to greatly simplify its formulation and the corresponding computation. However, continue to say that naïve Bayes offers optimal accuracy is far from the truth, because data from real life (such as the high correlation between fever and temperature in the study samples, reported in line 283) often violates the (naïve) assumption of independent predictors.
In line 242, notations in the formula are not in sync with those in the description, for example, p(x1|y) and p(y|x1) in line 244 never even appeared in the formula. The formula probably should have been:
P(y|x1, …, x7) = [P(y, x1) * … * P(y, x7)] / [P(x1) * … * P(x7)] = P(y|x1) * … * P(y|x7)
Also, the author uses symbol “y” to denote the outcome (rotavirus test positive), while in the early section use (y1…y6) to denote predictors of a data instance, please consider using consistent notations throughout the manuscript.
K-fold cross-validation
In line 258 it is stated that train, testing, and validation took 80%, 20%, and 20% of the total sample (n=509), which would add up to 120%. Perhaps, the authors were meant to say 80% for training, 20% for testing, and among the training samples, 20% (that is, 80% * 20% = 16% of the all samples by size) took turns to act as 5-fold cross-validation samples for various methods. Please clarify this abnormally.
Typically, simpler methods like linear and logistic regression do not require cross-validation, but methods with tuning parameters (also called hyper-parameters) such as the “k” of K-nearest neighbors, the maximum growth of the decision tree and random forest method, require cross-validation to find the best turning parameter.
In line 263, however, it seems the authors managed to apply a universal cross-validation procedure for all methods and computed the cross-validation scores, including linear regression and logistic regression which normally do not require cross-validation. If this is the case, please clarify that all methods were wrapped by a cross-validation procedure and the software package used to perform such task.
Results
In line 269, please change “children with …” to “children of …”.
In line 271, please change “children’s” to “children”.
In line 272, to specify a period in time, please use “between … and …” or “from … to …”, instead of “between … to …”.
The line 280 actually meant to measure the correlation among all variables involved, both the predictors and the outcome, including 7 clinical symptoms and possible infection associated with diarrhea.
From line 280 to 284, the authors described how to measure the correlation among the variables (clinical symptoms and outcome), but never mentioned the purpose of such measurement in the introduction or method section. As for the predictors, perhaps, the study is trying to avoid collinearity caused by highly redundant predictors, such as blood pressure and hypertension in a cardiovascular disease screening test. Collinearity cause model instability for many methods, and thus, it is crucial to drop all but one highly correlated predictor. For this study, the highest correlation (0.61) found between fever and temperature is still acceptable and both were kept as predictors for rotavirus infection.
Still, the purpose of including the outcome into the correlation coefficient measurement is unclear and I could make a guess about it. Please clarify the purpose behind the measuring of correlation among 7 predictors and the outcome.
In addition, the purpose and usage of correlation coefficient should be stated in the method section. In the result section, only the selected correlation coefficients, such as the one between fever and temperature (0.61) should be reported.
From line 289 to 291, these passages are making a subjective assessment of the methods, thus should be moved to the discussion section. If such assessment has been in the discussion, then these lines should be dropped. As a rule of thumb, the result section should focus on reporting results rather than making suggestions such as “method A is capable of performing task X better than method C”, while it is fine to report that “the accuracy of A is 3.4% higher than method C”, or simply lay out the accuracies of a few methods like the authors did between line 287 and 288.
From line 292 and 297, the introduction of ACU should be moved to method section, may be under a subsection named “performance measurements”, along with other metrics such as sensitivity, specificity, the confusion matrices, etc. Here in the results section, one should only report the AUC values the best performed methods.
Between line 295 and 297, the explanation of AUC is incorrect. An AUC close to 50% means the model prediction is as good (or as bad) as make a blind guess from population mean alone. (i.e., the prevalence of rotavirus infection among diarrhea children). In the study, the prevalence of infection is 365 / 509 = 71.7%, thus, blindly stating that every child with diarrhea is infected with rotavirus is likely to be correct 71.7% of the times. A prediction accuracy above but close to 71.7% will result in an AUC close to 50%, meaning the model is only slight better than blind guess. A prediction accuracy below 71.7% will results in an AUC below 50%, meaning the model have “deliberately” made predictions poorer than blind guess, which could happen when the outcome was mistakenly coded as 0=positive/infected and 1=negative/non-infected, or the samples in testing data drastically differ from those in the training data.
Moving onto line 298, my subjective judgment is that an AUC of 64% may be good enough, but 53% is bad. Again, these lines should be moved to the discussion.
From line 299 to 314, it would be better to refrain from lengthy explanation of the confusion matrix. Instead, the accuracy, AUC, and Macro F1 Scores summarized from a confusion matrix are much easier for the readers’ judgement. Of course, the full metrics of predictive performance should be kept but, in the appendix, or supplement materials. Also, it would be wiser to avoid explaining the confusion matrix in the method section, which is unnecessary for readers of biostatistics background but tedious for clinicians and public health workers.
Discussion
Between line 319 and 320, how come the deaths contributed by diarrheal (200,000) be less than the children killed by Diarrhoeal disease (525,000)?
Between line 325 and 327, a better expression would be “Several African countries have conducted studies to investigated the impact of routine rotavirus vaccination, but the results revealed that the rotavirus-associated deaths and hospitalizations remains unchanged within 2-3 years post-vaccine introduction”.
In line 332, the author shifted the topic from vaccine efficacy to machine learning, it would be better to start a new paragraph.
In line 335, a better expression would be “to better measure the effectiveness of routine vaccination…”.
In line 350, the selection bias typically refers to the poor choice of study samples that failed to represent the general population, rendering the results (i.e., the machine learning models) based on the study sample (503 children) non-generalizable to the greater public. Apparently, the next line talked about the very issue of poor generalizability, but using more methods will not solve the selection bias rooted in the study samples. What do the author really mean by “selection bias”? Could it actually be overfitted models? Please clarify.
Between line 358 and 360, it is meaningless to refer to performances of the Uganda study, unless the methods and models in the current study significantly out-perform the former, or the author want to explain why the current study could not achieve an accuracy as high as 100% reported by the Uganda study. Maybe these two lines should be put to the beginning of the discussion to justify the use of machine learning in this study.
From line 374 to 378, these past successes of machine learning in diagnosing other diseases should also be put forward as a justification of applying 8 machine learning methods in the current study.
Comments on Tables and Figures
Fig.1
By reading figure 1, one may guess the reason of the authors calculating the correlation among predictors is to check collinearity, which should have been clearly stated in the main text’s method section. Still, the reason to include the outcome into the correlation is unclear.
There is no sign if another 20% samples allocated to cross-validation. As suspected, is was probably the built-in mechanism of some methods (e.g., SVM, KNN, Forest, Tree, and XGBoost) for searching the best tuning (or hyper-) parameters, if that is the case, the main text should correct the wrong expression of 80% training, 20% testing, and another 20% for validation (which adds up to 120% of the study samples).
The second last box (in dark blue) should be “Performance Evaluation”, instead of “Prediction Model”.
Fig. 2
(Please refer to the first comment for fig.1, regarding the use of correlation coefficient)
Fig. 3
(No comments)
Fig. 4
The AUC for Random Forest (RF) and SVM is less than 50%, despite similar accuracy performance in the testing data sets in comparison to other methods, especially between the Random Forest and Decision Tree. When a predictive model results in an AUC < 50% (a.k.a., deliberately making wrong prediction), one could simply reverse the predicted come to achieve and AUC > 50%.
Could it be that the RF and SVM automatically coded the minor outcome (29% non-infection) as 1 and major outcome (71% infection) as 0? Some software automatically assumes minor outcome as disease cases and the major outcome as (relatively) healthy controls; or it could be that the sample characteristic of training and testing data are drastically different, a few re-runs with a different partition should rule out this possibility.
Please investigate why the AUC went below 50%, and make sure to the AUC is always equal or greater than 50%.
Fig.5
Two major issue concerning the study design are identified and must be fixed.
First, the ratio of cases (infection) versus controls (non-infection) are drastically different between the training and testing samples. For example, based on Fig.5a, the case ratio the training samples is (41+83) / (41 + 83 + 8 + 275) = 0.305, but in the testing samples, the ratio is (3 + 7) / (3 + 7 + 17 + 75) = 0.098, where the latter is quite different form the former. This is possibly the reason why some model reported AUC < 50%.
Second, the assignment of training and testing samples are not consistent across different method. To continue with the example, based on Fig.5a, the number of cases and controls in the training data of decision tree were 41+83 = 124 and 8 + 275 = 293, respectively; based on Fig.5b, however, the number for XGBoost were 66 + 54 = 120 and 15 + 272 = 287, respectively. Therefore, two methods were given different training (and testing) data. As a consequence, the comparison of prediction performance was invalid, since the model built by each method was based on a different “playing field”.
The authors must fix these issues and re-run the analysis, that is, the ratio of infected and non-infected children should be the same between training and testing data (both reflect the overall prevalence o infection in the full data); and, all 8 methods should be supplied with the same pre-divided training and testing data in order to make sound judgment of their relative performance.
Table 1.
The temperature predictor is a 3-level categorical variable, according to the code book, it is coded as 3 indicator variables, however, one of them would be redundant (since one can always workout the third knowing the other 2) please ensure it is coded as two dummy variables, where normal may serves as the reference level (0, 0), mild is coded as (1, 0), and moderate as (0, 1).
Table 3.
First, please clarify that the scores for performance metrics other than accuracy are based on the testing data sets.
As for accuracy, it is unnecessary to report the score based on training data. If the authors are concerned about overfitting, where performance in the training data would be (unreasonably) high even if the performance in testing data is poor, the pairs of bars in Fig.3 has provided that information.
For an imbalanced sample (having much more infection than non-infection), the (Macro) F1 score is actually a better metric for prediction performance, and should be reported in the main text in place of precision, recall, and specificity.
The accuracy, which is also call Micro F1 Score, can still be reported in the main text, along with AUC.

Reviewer 2 ·

Basic reporting

no comment

Experimental design

While the use of regression-based algorithms, such as linear regression, for the classification problem raises questions about their applicability, the rationale behind these choices could be further elucidated for the reader's understanding.

Given that the models discussed have attained classic status within the field, elaborating on their general principles offers limited new insights. Instead, the manuscript could significantly benefit from a detailed exploration of the model development process. This includes the hyper-parameter optimization techniques employed, strategies implemented to mitigate overfitting, and the rationale behind the selection of features. Such information would not only augment the paper's methodological rigor but also provide valuable practical insights into the model's construction and evaluation.

Validity of the findings

The AUC values presented in Figure 4, notably 0.53 and 0.64, suggest there might be potential for enhancing the model's predictive accuracy. An AUC of 0.53 approximates the performance of a random classifier, and while 0.64 indicates some predictive capability, it will benefit from further optimization to achieve more robust modeling results that would be compelling to the broader scientific community.

Additionally, there seems to be a discrepancy between the reported Recall/Specificity values and those depicted by the ROC curves. Such differences warrant clarification; if the metrics presented are based on the training set, this should be explicitly stated. Incorporating test set metrics is also crucial, as they provide a critical assessment of the model's generalizability and can highlight potential overfitting issues.

In the abstract results section, the authors quote precision and recall for separate models without any reference to other metrics: this does not provide robust information about the quality of the models as other metrics are not quoted for the thresholds chosen.

Given these points, I encourage a thorough re-evaluation and more detailed presentation of the findings in case they have reasonable predictive power on the test sets.

Reviewer 3 ·

Basic reporting

Your groundbreaking research on developing machine learning models to predict rotavirus infection in children within resource-limited settings is truly remarkable and well-timed. By employing supervised learning techniques to address this critical healthcare issue, you've exemplified a proactive approach toward enhancing pediatric healthcare outcomes. Before we delve deeper, I want to acknowledge the significance of your work in advancing our understanding of rotavirus-associated acute gastroenteritis (AGE) diagnosis, particularly in regions where access to conventional diagnostic methods may be limited. I have a few questions which I believe to add more value to the readers and use the principle unanimously.

Experimental design

1. How were potential confounding factors, such as demographic variables or comorbidities, considered in the development of the machine learning models to ensure the predictive accuracy of rotavirus infection in children?
2. Given the moderate AUC values of the decision tree and XGBoost models, what specific features or aspects of the dataset might have posed challenges for achieving higher predictive performance, and how could these challenges be addressed in future iterations of the models?
3. The study employed correlation analysis to explore the relationship between clinical symptoms and rotavirus infection. Can you discuss any unexpected correlations or insights gained from this analysis that could inform the understanding of rotavirus pathogenesis or clinical presentation?
4. False positives and false negatives in the confusion matrices indicate potential areas for improvement in model performance. How might the models be refined or augmented to reduce these misclassifications, particularly in distinguishing between rotavirus-positive and -negative cases with similar symptomatology?

Validity of the findings

5. Integrating machine learning models into health information systems requires careful consideration of data privacy, scalability, and usability. What strategies or frameworks could be adopted to ensure the ethical and practical deployment of these models in resource-limited healthcare settings, while also addressing potential biases and disparities in access to healthcare services?

Additional comments

No

---

## Round 0.2 · Major Revisions

Thank you for submitting a revised version of this manuscript. We recognize that many of the reviewer comments have been at least partially addressed in the current version. However, the reviewers' point out some points that have not yet been adequately addressed. We recommend that you thoroughly address the current (and prior) reviewers' comments in order to move this manuscript forward. It is possible that this could require you to update and/or redo a subset of the analyses and the associated results.

·

Basic reporting

The structure and writing should be revised to deliver the methods and results accurately and concisely.
(1) The method section did not provide a “Feature Selection” sub-section.
(2) multiple instances of redundant writing.
(3) Mislabeled figure and difficult read text in the figures.
(4) Results should be concise and focus on reporting the results only.
(5) (suggestion) downsize the reporting of full-feature analysis, focus on selected-feature analysis.
(6) the use of 10-fold cross-validation and the overall 80%/20% sample split are still confusing.

Experimental design

Asides from the two major flaws addressed so far, there are some remaining issue in the study design.
(1) The use linear regression is still problematic
(2) suggest: treating feature selection as a common, mandatory procedure, not for a comparison with full featured analysis.

Validity of the findings

No comment

Additional comments

Thanks for addressing most of the many syntax and technical issues listed in the previous review, especially major flaws in the study design.
The current review can focus more on the general design, structure of the writing, and less so on line by line comments.
I did not go over the discussion section because I believe large part of the explanation and interpretation within the results section should be moved into discussion.
Although issues in the study design are mostly addressed, multiple revisions are still required to improve the delivery of message in a accurate and concise manner.
[Study Design]
(1) The use linear regression is still problematic because the predicted outcome is a continues value while the actual outcome (infection) is binary. It is therefore unclear how the author managed to calculated all the performance metrics except AUC. Ironically enough, Figure 3 did not report the AUC of linear regression. The authors could consider removing linear regression from the methods all together.
(2) I suggest treating feature selection as a common, mandatory procedure, not for a comparison with full featured analysis.
[Structure and writing]
(1) The method section did not provide a “Feature Selection” sub-section to cover Correlation Analysis and ANOVA. Instead, the rationale and reason behind the Correlation Analysis and ANOVA were covered in the results selection.
(2) There are multiple instances of redundant writing, see comments on [L255-260] and [L268-272].
(3) Mislabeled Figure(s), see comments on [L321-321], figure 1, and supplement figure 1; the labels and text in figures should be enlarged to help reading.
(4) Results should be concise and focus on reporting the results only, not trying to explain why certain algorithms is good or bad. The explanation of good or poor performance and comparison with previous studies should move to the discussion section.
For example, lines like “The reason behind the improved accuracy achieved by RF is that it does not require normalization or scaling of the data, avoids overfitting, and is good at discovering patterns from complex medical datasets.” or “A possible reason behind poor classification results is that the NB classifier performs well when the features are independent, but in this dataset, we have already observed from our correlation analysis that most of the features seem equally important which is the ideal case for DT to work well.” should be moved to the the discussion.
(5) Instead of detailed reporting of both full and selected feature analysis, consider a simplified reporting focusing on selected features only, while briefly mention that the full feature version showed similar trend (i.e., RF was best performing) and similar performance.
Feature selection is already a common procedure in machine learning, and the focus of the manuscript are the comparison among 8 algorithms, not the difference between full and selected feature analysis. Therefore, unless the full feature analysis showed drastically better performance among the most promising methods (i.e., RF and XGBoost), feature selection should be treated as a common procedure, and the full featured analysis is worthy of a separated reporting.
(6) The manuscript still hint the performance was measure by averaging the 10 results from 10-fold cross-validation (CV) with a 90%/10% split, which confuses with the one time 80%/20% sample split.
See the comment on [L054-055] [and L198-199] as well.
[Line by line comments]:
L046 -057: results in the abstract should be brief, (1) report one best performing algorithms (seem to be RF) of your recommendation and the feature space used, after balancing different performance measures; (2) there is no need to run down every performance metrics - the F scores and AUC which are more informative.
L049-050: F2 score was not reported for RF.
L053-056: how come XGBoost and DT were also the best since RF was reportedly already the best of all?
L054-055: the writing “average area under the curve” sound different from the ordinary “area under the curve”, why would a model have multiple AUC values to average from, if there was only one set of testing sample (i.e., the 80% training /20% testing split)? Was it an error that the authors reported the average of 10-fold CV’s internal performance metrics (based on ten times of 90%/10% splits among the 80% training data)?
L198-199: are you sure the K-fold cross-validation was not used to evaluate the models? As discussed in the previous review, the cross-validation is used for training only (i.e., to determine the hyper-parameters), and the evaluation is performed on the 20% testing data.
L200-242: when describing the 8 ML methods, there is no need to include details of the formula, but it should be broken into 8 paragraphs to help the reading.
L212-212: the abbreviation CNN was never introduced
L255-260: the listing of 8 performance metrics were repeated twice.
L268-272: the description of ROC and AUC is redundant because they were re-introduced in greater detail by sentences immediately after.
L276-L281: I wrote these in the previous review to explain why AUC < 0.5 is erroneous, so the authors could correct their calculation (to ensure AUC never go below 0.5), it was not intended to be copy-pasted as part of your explanation of AUC!
L321-321: Figure 2b was not provided.
Figure 2b: it would be better to replace the x and y axis labels from feature 1-6 to actual variable names like those in 2a; also the labels should be enlarged to help reading.
Supplement Figure 1: the labels should not be sensitivity / specificity, but like the previous version.

Reviewer 2 ·

Basic reporting

No comments

Experimental design

The paper states that SVM is designed to handle non-linear relationships. While this is generally true, which kernel is used in the paper? using the linear kernel still results in linear relationship modeling.

While the authors present a schema of train/test datasets split, I am wondering how the test data was used in preprocessing. I.e., was any test data used in deriving the missing values/feature design? If so, the recommendation would be to only use train data for these purposes to avoid data leakage.

Validity of the findings

I have a question regarding the ROC curve shapes for the models that were reported as the best performers. Specifically, I noticed that the ROC curves, such as the one for the Random Forest model, appear to have only a single point connecting two straight lines. This is unusual because the model should output continuous risk scores, which typically result in a smoother ROC curve reflecting various thresholds and their corresponding false positive rates (FPRs) and true positive rates (TPRs). I would suggest using continuous scoring for the ROC curve demonstration, as this is a common practice and provides a more accurate representation of the model's performance across different thresholds.

---

## Round 0.3 · Minor Revisions

I would strongly encourage you to adequately address the comments provided by both reviewers.

·

Basic reporting

The manuscript has seen improvement in narrative structure. Redundant writing still exists but are much less prevalent then previous revisions. There are still issues in English grammar, I suggest using online AI assistance such as “Grammaly” (www.grammarly.com) to further improve the grammar.

As the quality of reporting improves, fewer issues were identified in method and results sections, listed in the “Line by Line Comment”.

There still design issue to be address to justify a publication. I do not go through the entire discussion before issues in the design and reporting are addressed.

Experimental design

[Possible design issues]:
(1) The authors reported high correlation between “Fever” and “Temperature”, also stated that they considered removing either of them to avoid collinearity; however, while both features were retained throughout the subsequent machine learning analysis, there is no explicit statement or reasons given regarding why both are retained, despite the stated concern of collinearity.
See the comments for L295-L302 for details.

(2) authors should make explicit statement that the 7 performance measures are calculated on the 20% testing data samples. If the measures were calculated on the 80% training samples (hopefully not), they have to be re-calculated and re-reported.

Validity of the findings

See [Possible design issues]

Additional comments

[Line by Line Comment]:
L289-L293: (1) although “Diarrhea duration (days)” is negatively correlated with “rotavirus”, reporting of the correlation should not be skipped, as L307-308 later stated that “Diarrhea duration have a significant influence in the prediction of rotavirus disease”; (2) apparently, feature “Dehydration” was dropped due to low F-Test score, however, its correlation with “rotavirus” should not be skipped both here, and in Figure 2(a); (3) when reporting correlations between the 7 clinical features (5 + Diarrhea duration + Dehydration) and the 1 decision feature (rotavirus) in the manuscript, ordering of the 7 correlation coefficients should align with those in Figure 2(a) to improve reader’s experience.
L295-L302: it is clear “Fever” and “Temperature” are highly correlated (0.67) and the authors considered removing one of them to avoid collinearity. However, a clear reason was not given to eventually retain both for the ML analysis. Perhaps, due to both having high F-test score? If the F-test were done 7 times on a per-feature basis instead of getting all 7 F-test scores in a single multi-variate test, then the high scores do not address the issue of collinearity; or, a correlation of 0.67 is simply not that high to justify the removal of one features? Please add an explicit statement that both “Fever” and “Temperature” despite their high correlation, and provide some reasoning in the discussion why both were retained. The best reasoning would be that removal of either significantly reduce the ML performance, but such reasoning would require more analysis (i.e., 7-features vs 6-features).
L310-L313: the authors cited prior study [58] in support of “Dehydration” as one of the “factors for rotavirus diagnosis”; immediately after however, they reported that “Dehydration” received a very low score in both correlation and FBFS. So, (1) apparently, “Dehydration” was excluded from ML due to the low scores, its exclusion should be explicitly stated here; (2) “Dehydration” received low scores despite citation [58] suggesting it as a predictor of rotavirus, the contradiction (i.e., why Dehydration is clinical relevant, but weakly associated with rotavirus?) was never acknowledged and addressed; suggested edit: “Nnukwu et al [58] have reported .…, and death. [Surprisingly/However], ‘Dehydration’ has received a very low score in both our correlation and FBFS analysis, [and is thus excluded from machine learning analysis] [see Discussion for possible explanations]”. It is not amendatory to explain the contradiction, but doing so in the discussion can strengthen the manuscript (please do not mechanically copy-paste what I suggested).
L320-L321: the training time is not clear. The number of iterations to reach convergence is different for different methods; it is better to report to total seconds spent by each methods to reach convergence on the 80% training samples.
L321-L323: is the performance metrics reported here and in Table 3 based on 80% training samples or 20% testing samples? Please make explicit statement that those measures were calculated with predicted retrvirus infection versus truth based on the 20% testing samples. If the authors reported the metrics based on training samples, the numbers has to be re-calculated and re-reported based the 20% testing samples instead. High predictive performance of the ML modeling on the training samples are necessary of course, but meaningless if the same models performed poorly on testing samples (i.e., a sign of over-fitting).
L330-332: since the comparison between 8-feature (no selection) and 7-features (“Dehydration” excluded) was no longer the main point of the manuscript and feature selection has been treated as a standard procedure, these lines are redundant and should be removed. If the authors wish to include the work involving all 8-features, I suggest the statement “a sensitivity analysis with all 8-features (including “Dehydration”) was also conducted, and the predictive performance of ML models on the same 20% testing sample are similar to select features (excluding “Dehydration”), see supplement table # for details”. So, in an addition supplement table, provide the same report as Table 3, based on all 8 features.
L332-225: discussion of feature scaling and the reasoning to not used it in the current study, should be moved to the Discussion section.
L366-L367: the correlation between temperature and rotavirus is only 0.039, lower than between vomiting episode per day and rotavirus which is 0.087; did the authors meant to say that the heatmap showed high correlation between temperature and fever which was 0.67?
[Figure 1]: For the same reason, please simplify figure 1 to only include the “data set without Dehydration”, and remove “data set with all features”.
[Figure 2a]: (1) make sure to report the correlations among all 7 clinical features plus the decision feature prior to feature selection (i.e., a 8x8 instead of 7x7 correlation matrix including “Dehydration”); (2) the caption and description of figure 2(a) does not clearly state what the figure truly represent, which is the correlations between 7 clinical features and rotavirus infection summarized from (N=?) children with diarrheic symptoms; I suggest to combine the caption and description into a caption alone, like “Figure 2(a) […]” (Note: when fixing the caption of Figure 2(a), please do not mechanically copy-paste what I wrote here).
[Figure Numbering]: apparently, PeerJ’s PDF compilation automatically numbered the figures as 1, 2, ..., so the figures 2(a), 2(b), figure 3 were re-numbered by PeerJ as Figure 2, 3, and 4 instead. Please work with PeerJ to fixed the numbering of figures.

Reviewer 2 ·

Basic reporting

Please double check for the imprecise wording/misprints across several sections. Two examples provided in the sentences below:

1. 'A comparison of the performances of the 7 models using the classification results obtained.' - missing 'was'.
2. 'All the 7 classification models such as RF, NB, DT, KNN, SVM, Logistic Regression, and XGboost were trained on the data with 80% training and 20% testing subsets. ' - they were only trained on the train data, the wording is slightly imprecise.

Experimental design

No comment

Validity of the findings

No comment

---

## Round 0.4 · accepted · Accept

I believe that the authors have sufficiently addressed the reviewers' previously-stated comments related to structure, results, etc.